# Disposal Behavior of Used Masks during the COVID-19 Pandemic in the Moroccan Community: Potential Environmental Impact

**DOI:** 10.3390/ijerph18084382

**Published:** 2021-04-20

**Authors:** Nezha Mejjad, El Khalil Cherif, Antonio Rodero, Dorota Anna Krawczyk, Jauad El Kharraz, Aniss Moumen, Mourad Laqbaqbi, Ahmed Fekri

**Affiliations:** 1Laboratory of Applied Geology, Geomatics, and Environment, Ben M’sik Faculty of Sciences, Casablanca 20670, Morocco; mejjadnezha@gmail.com (N.M.); ahmedfekri13@gmail.com (A.F.); 2Department of Geology Laboratory of GMSSURAC 45, Faculty of Sciences, Chouaib Doukkali University, El Jadida 24000, Morocco; 3Laboratory of Chemistry Research Unit (CIQUP), Faculty of Sciences, University of Porto, 4099-002 Porto, Portugal; 4Institute for Systems and Robotics, Instituto Superior Técnico, University of Lisbon, 1649-004 Lisbon, Portugal; 5School of Engineering Sciences of Belmez, University of Cordoba, 14071 Córdoba, Spain; 6Faculty of Civil Engineering and Environmental Sciences, Bialystok University of Technology, 15-351 Bialystok, Poland; d.krawczyk@pb.edu.pl; 7Global Change Unit, University of Valencia, 46010 Valencia, Spain; jauad@uv.es; 8National School of Applied Sciences of Kenitra, University of Ibn Tofail, Kenitra 14000, Morocco; aniss.oumoumen@uit.ac.ma; 9Laboratory of Advanced Materials and Process Engineering, Multidisciplinary Faculty Of Sidi Kacem, Ibn Tofail University, Kenitra 14000, Morocco; mouradlaqbaqbi@gmail.com

**Keywords:** COVID-19, behavior, facemasks, Morocco, environmental impact, plastic pollution, public health

## Abstract

The spread of coronavirus disease-2019 (COVID-19) levied on the Moroccan authorities to increase their mask production capacity, which reached up to 12 million facemask units produced per day. This increase in personal protective equipment (PPE) production and consumption is an efficient tool to address the spread of COVID-19. However, this results in more plastic and microplastic debris being added into the land and marine environments, which will harm the ecosystem, wildlife, and public health. Such a situation needs deep individual behavior observation and tracking, as well as an assessment of the potential environmental impact of this new type of waste. For this reason, we assessed the Moroccan population’s behavior regarding the use and disposal of facemasks and gloves. An exploratory survey was prepared and shared via social media and email with the population of Rabat-Salé-Kénitra and Casablanca-Settat regions. Additionally, we calculated the estimated number and weight of daily and weekly PPE used and generated by the studied regions. The survey showed that 70% of the respondents threw their discarded masks and gloves in house trash or trash bins after their first use, whereas nearly 30% of respondents admitted that they did not wear masks because they did not leave their homes during the lockdown, while from the 70% of facemask users, more than five million (equivalent to 40,000 kg) of facemasks would be generated and disposed of daily by the community of these regions, which presents 35% of the total engendered facemask waste in Morocco. Accordingly, the environment impact of facemasks showed that the greenhouse gas footprint is about 640 kT CO_2_ eq./year for the whole of Morocco, while the energy footprint is around 60,000 GWh/year. Furthermore, an urgent multidisciplinary environmental assessment of the potential impact of PPE must be conducted among the 12 Moroccan regions. This study demonstrated the real impact of the COVID-19 PPE on human behavior and the environment and suggests a need for providing new didactic management of facemasks and gloves.

## 1. Introduction

Following the recommendations of the public health emergency of international concern (PHEIC), the World Health Organization (WHO) declared COVID-19 as a global pandemic on 31 January 2020 [1,2]. COVID-19 has infected in 31 January 2021, more than 102 million of the world population and caused 2,211,762 deaths (WHO 2021). COVID-19 can be transmitted through physical contact and respiratory droplets, which are generated when an infected individual sneezes or coughs. To date, the Severe Acute Respiratory Syndrome (SARS) and Middle East Respiratory Syndrome (MERS) transmission modes are different, and coronaviruses could remain viable in the environment for hours and even days and could be a source of transmission [3,4,5,6]. However, COVID-19 could be transmitted through touching the objects or surfaces where the virus occurs, and then re-touching the eyes, mouth, or nose [7,8,9].

To reduce and limit the spread of COVID-19, numerous countries around the world enforced confinement and obliged the use of masks for active individuals as a preventive measure. This practice was imposed in China and Hong Kong during the SARS epidemic in 2003 by catering companies to protect workers and the public [10]. Currently, the trade in medical products has risen extremely, specifically for products related to the prevention, testing, and treatment to fight against COVID-19 [11].

In Morocco, the first COVID-19 case was announced on 2 March 2020. By 6 April 2020, the number of confirmed cases had reached 1120 with 80 deaths, which led the government to impose the wearing of facemasks in all workplaces and public spaces on 7 April 2020. Since then, all Moroccans have been obligated to use facemasks when they leave their houses for any reason (groceries, work, etc.). This situation has increased the demand for personal protective equipment (PPE) such as facemasks and gloves, while many countries were temporarily banned from the export of protective masks due to the COVID-19 pandemic [11]. Under this critical situation, the Moroccan authorities mobilized a group of manufacturers to contribute to the local production of protective masks (non-woven fabric) for the local market. Currently, 12 million units per day are produced.

Notably, this increase in the production and consumption of personnel protective equipment items across the world to fight against the spread of coronavirus adds to the already existing concern over plastic pollution and its effects on the environment and wildlife if these items are not disposed of properly. The occurrence of diverse types and colors of facemasks in an ocean in Hong Kong was reported in February 2020 by an organization dedicated to marine pollution researchers, named Oceans Asia [12]. In Peru, about 138 facemasks associated with the COVID-19 pandemic were found in eleven beaches located along the coast of Lima during 12 sampling weeks [13]. Other investigations based on photographs and videos have also shown the first evidence of PPE pollution in different environments, including coastal areas such as those presented in [14,15,16]. In Morocco, survey results of a study conducted in two cities (Khenifra and Tighassaline) showed that almost all respondents disposed of their used PPE items in the same trash bin with household waste [17]. Thus, these PPE items would end up in nature and easily reach the marine environment in absence of a strategy related to such waste disposal practices and management.

Generally, most COVID-19 studies have focused on its causes, origin, and its potential impact on water and air quality [18,19,20,21,22], while few of them have documented the possible personal protective equipment waste impact on the environment and the management strategy and approaches to address facemask waste disposal [23,24]. In this sense, recent studies have highlighted the urgent need for developing new management approaches to this kind of waste to avoid the environmental harm and health risks that could be caused by the disposal of facemasks and other PPE items [25]. Numerous studies have proposed solutions to tackle the increase in PPE waste and minimize their long-term impact on the environment and wildlife. Xiang et al., 2020 [26], have proposed the decontamination of surgical facemasks and N95 respirators through using a dry heat pasteurization method which retains their filtering capacity, as a solution for the facemask shortage during the early COVID-19 period, which could also help to reduce the quantity of generated facemasks. Among the environmentally friendly approaches that would help in resolving the facemask waste issue, we cite the conversion of N95 mask waste to steam and electricity by combining power plant and heat, and to energy-dense oil products through a hydrothermal liquefaction process, and to ethanol by using a syngas fermentation method [27]. Turning discarded facemasks into renewable liquid fuels by converting polypropylene (plastic), which is the raw material of masks, into biofuel, is a strategy suggested by [28] to mitigate the problem of discarded facemasks associated with COVID-19.

The present study aims at assessing the Moroccan community behavior towards facemasks and gloves, as a new custom in the fight against COVID-19, which seems to be an adequate way to investigate the potential environmental impacts of PPE. Accordingly, this study links the community behavior toward the use and management of PPE item waste, and the resulting environmental impacts from mismanaging and littering these items in streets, and finally raises the question around the long-term impact of this new practice through estimating the quantity and weight of daily and weekly used facemasks by the Moroccan community.

## 2. The Possible Impact of Used PPE

About 12 million units per day of facemasks (non-woven facemasks) are produced by Moroccan factories and distributed all around Moroccan cities to limit the spread of COVID-19 [29]. The continuous spread of COVID-19, even with the imposed lockdown, led the Moroccan authorities to enforce the public to wear facemasks in the workplace and public spaces to avoid the transmission of the COVID-19 and control its spread. Despite the multiple recommendations and guidance released by the media to warn people about how to use and remove these facemasks and to dump the used ones directly in a closed bin, the used facemasks have been littered everywhere, in roads, gardens, hospitals, beaches, etc. (Figure 1). These discarded facemasks could contribute to the spread of the virus. According to [30], once 80% of the population wears a mask, the influenza outbreak can be stopped immediately. Nevertheless, if this 80% disposes of their used masks on roads, this could create a new form of spread of the COVID-19 virus. To the best of our knowledge, few analytical or experimental studies have examined the effects of dumping used masks or gloves by the general population on roads on human health, or on the environment. All the previous studies have focused on the management of hospital waste, including gloves, masks, and headcovers [31,32,33,34,35]. This kind of waste could be hazardous if it is misused and/or mismanaged. The possible impacts of throwing PPE waste without treatment out in nature and on the road are presented in the flowchart below (Figure 2). 

Moreover, the method of using the facemasks could contribute to COVID-19 transmission. A study was conducted from 28 October 2014 to 31 March 2015 on health care personnel to assess the risk of contamination of their skin and clothing. The latter revealed that despite the use of PPE, the risk is especially high during the removal of PPE [36]. This study might indicate that the use of gloves and facemasks by the public and the incorrect removal of this PPE could present more risk to them, especially if they are in close contact with infected persons. Thus, discarding this PPE in an open bin may also increase the risk of transmitting the virus to sanitary workers and/or stray animals, because COVID-19 could persist on the outside of a surgical mask for seven days [37]. In Morocco, few studies have been carried out to evaluate the possible risk of medical waste [32,34,35]; these studies focused on hospital wastewater management, while about 21,000 (2010) tons per year were left behind by medical and pharmaceutical waste [38]. A total of 11,910 tons per year of medical waste from public and private health establishments were reported in WASTESUM (LIFE06 TCY/MA/000254), 2010. Consequently, this estimated number will considerably increase with the increasing rate of the use of personal protective equipment by health care workers as well as the community. According to El-Ogri et al., 2015 [34], the pollutants generated from hospital services are discharged into the sewerage system without any treatment except for liquid waste engendered by laboratories and operating suites, which could harm the environment and human health. 

## 3. Materials and Methods

### 3.1. Study Design

We describe and analyze the impact of changes caused by the fast spread of COVID-19 regarding the use of protective facemasks and gloves around the world, especially in two regions: Rabat-Salé-Kénitra and Casablanca-Settat, Morocco (Figure 3). In these regions, we investigated the populations’ behaviors towards the use and disposal of the facemasks and gloves during the COVID-19 pandemic and their possible effect on human health and the environment. Furthermore, these two regions are known for the concentration of the most important factories and industrial units in the north of Morocco [39], together with the Tangier-Tétouan-Al Hoceima region [40].

The use of facemasks is a new culture for the Moroccan population, unlike in some Asian countries [41,42] where people are used to wearing them because they have already experienced such epidemics, and because of heavy traffic and seasonal air pollution. In this sense, the study analyzes and discusses the community behavior toward the use and management of PPE during the COVID-19 pandemic. Understanding the ways facemask are disposed would help in decision-making regarding community management and in establishing and developing sustainable waste strategies. 

### 3.2. Data: Population and Survey Questionnaire

An online survey was performed between 19 and 23 May 2020 to investigate Moroccan behavior towards PPE (facemasks and gloves) and to determine the impacts on the environment if they are disposed of improperly. Due to the quarantine measure taken by Moroccan authorities, it was impossible to take a survey questionnaire in the field. Therefore, to avoid any possible contact with infected people, this survey was answered by the Moroccan community through social media channels (Facebook, Twitter, and WhatsApp) and email. The exploratory survey rendered 185 responses from individuals aged over 24 years old. It should be be noted that 45% of Moroccan Facebook users are older than 24 years and are living in the Casablanca-Settat and Rabat-Salé-Kénitra regions [43].

However, the survey was composed of the following parts:(i)First part concerned the general information of participants (gender, age, city, education level, and employment sector);(ii)Second part was related to the behavior and habits during COVID-19 regarding the respect of lockdown and disinfection;(iii)Third part was focused on the frequency of use of facemasks and gloves;(iv)The last part was dedicated to the management of discarded facemasks and gloves. All the participants were informed about the confidentiality of their responses as well as the purpose of conducting this exploratory survey.

### 3.3. Statistical Analysis and Reliability Analysis

The statistical analysis was executed by using the statistical software package SPSS (SPSS Inc. 1Released 2007. Version 16.0. SPSS Inc., Chicago, IL, USA).

In the first step, we conducted a reliability analysis of the survey using Cronbach’s alpha, which is an easy method that can be used to ascertain the internal consistency of the items of the survey, according to Tavakol & Dennick, 2011 [44]. The calculated alpha values were 0.6, acceptable for the no-high question number of this survey. 

To determine the correlation of each survey item with the total survey score and delete the item without correlation, the factor analysis technique was used with principal component analysis (PCA) as the factor extraction method. The Kaiser–Meyer–Olkin (KMO) measure of sampling adequacy is a factor that confirms if the sample is sufficient for factor analysis. A second statistical value that confirms the sampling adequacy is Bartlett’s test of sphericity, which confirms if the correlations of all items are different from zero. The values of the KMO index and Bartlett Test were 0.7 and 0.001 (<5%), respectively, which allowed us to pursue the frequency analysis about the agreement and disagreement levels of all individuals [44].

Another aspect to be considered in the reliability analysis is the statistical validity of the size of the sample set. As mentioned above, the number of responses was 185. Considering the relationship between the size of the sample set, the margin of error, and the confidence level for an unknown population:(1)$$N_{\mathrm{SS}}=\frac{Z^{2}\sigma^{2}}{e^{2}}$$
where $N_{\mathrm{SS}}$ is the necessary sample size, Z is the Z-score, which depends on the confidence level (Z = 1.645 for 90%), e is the margin of error, and σ is the standard deviation of the population (in normal conditions: 0.5).

In our sample set, for an acceptable confidence level of 90%, the margin of error was only 7%.

A summary of the statistical parameters corresponding to our study is presented in Table 1.

## 4. Results

### 4.1. Moroccan Community Behavior and Habits during the Confinement

The survey questionnaire focused on analyzing the behavior and habits of a part of the Moroccan community living in the cities located between Kénitra and El Jadida where important industrial activities are concentrated.

The obtained results from the exploratory survey questionnaire related to the behaviors and habits of the community of Casablanca-Settat and Rabat-Salé-Kénitra regions during the confinement showed that 30% of participants were respecting the lockdown and almost all contributors followed sanitary measures recommended by the Ministry of Health. A minority of individuals behaved differently and did not consider the health ministry recommendations (Figure 4). Figure 5 indicates that the individuals who did not respect sanitary measures did not leave their houses and were under lockdown. Nevertheless, this population sample was almost behaving positively towards the recommended prevention measures and the level of awareness amongst them was practically high.

### 4.2. The Use of Personal Protective Equipment (PPE)

The analysis of the usage rate of facemasks and gloves within the individual samples revealed that more than 70% of the persons used masks once, twice, or more than two times per day (Figure 6). However, the share of respondents regarding the use of gloves to protect from COVID-19 revealed that only 3% wore gloves. In comparison with other countries, an online survey conducted in Europe between 22 and 28 June 2020 about the use frequency of facemasks has shown a colossal difference in terms of wearing facemasks during the coronavirus pandemic [45]. About 84% of Spanish and 83% of Italian respondents always wore facemasks, while only 3.6% from Norway, 2.4% from Sweden, 1.9% from Finland, and 1.6% from Denmark always wore facemasks outside to protect from COVID-19. The difference between each country legislations regarding facemask wearing is the main reason for the variation of countries’ communities’ behavior over the wearing of masks in public during the COVID-19 pandemic.

A recent study [46] showed that if 70% of the Moroccan population wear at least one mask per day, the estimated number of facemasks generated and disposed of daily will be around 16,537,438, equivalent to 140,568 kg of facemasks that could be disposed daily in different environments by people living in Morocco. In the same way, we estimated the total daily generated and disposed facemasks for the two studied regions, and surveyed the population using the following equation [46,47]:(2)$$Total\;generated\;facemasks\;per\;day=\frac{\left(Tp\times Up\times Ar\times Ac\right)}{10000}$$
where *Tp* is the estimated population for each region (Rabat-Salé-Kénitra; Casablanca-Settat, and surveyed population in the present study), *Up* is the percentage of the urban population in the two regions, *Ar* = 70% is the obtained percentage rate of facemask and gloves acceptance in the present study, while *Ac* is facemasks by capita.

According to our estimations, if the population of two studied regions wears at least one facemask daily at the rate of 70% facemask acceptance, more than five million facemasks would be daily generated and disposed (Table 2). Considering the average weight of a facemask, which is equivalent to 8.58 g [46], the total weight of facemasks used daily in the two studied regions would exceed 40,000 kg. This quantity of discarded facemasks from people living in those regions presents approx. 35% of the total generated and disposed facemasks in the whole country. This means that with around 41 million facemasks discarded weekly, over 345 tons of facemasks waste would be generated each week and littered or improperly disposed of by people living in these regions. Considering the daily produced municipal solid waste in the two regions, which is equivalent to 8.7 million kg (0.76 kg/capita) [48], the resultant waste from discarding facemasks presents 0.46% of the total daily waste generated in Morocco.

It should be noted that from seven million tons of solid waste generated yearly in Morocco, 572,000 tons of plastic waste are produced [50], which represents around 753 tons of plastic waste generated daily. Accordingly, the 40,000 kg of daily generated waste of facemasks represents 5% of the total daily produced plastic waste in the country. This kind of waste is manufactured with non-woven materials, which include polypropylene [46]. These plastic products could end up in nature in different ways if they are thrown on roads and streets. It was reported that marine animals such as seabirds and turtles could easily mistake plastic for food [51] including facemasks owing to their bright colors and soft construction. 

However, the rate of wearing facemasks has increased between April and early June, as shown in Figure 7. In comparison with European countries, the United States of America, and China, the use of facemasks has considerably increased during the COVID-19 pandemic and reached 90%, 52%, and 20% in the United States, Germany, and the United Kingdom, respectively. This positive trend regarding wearing facemasks in the fight against the COVID-19 is mainly related to the upward trend of COVID-19 cases between April and June. In Morocco, the High Commission for Planning (HCP) announced that almost all Moroccans must have facemasks to protect themselves from COVID-19, and the proportion of Moroccan households with protective facemasks reached 97% by June (99% in urban areas, and 93% in rural areas). These data released by HCP is in good accordance with the survey results.

In order to understand why a percentage of respondents do not wear facemasks, we analyzed the relationship between being outside and wearing masks (Figure 8). The results showed a positive correlation between being outside for any justified reason (groceries, work, etc.) and wearing facemasks. Thus, only people leaving their houses wear facemasks as a protective tool against COVID-19.

Figure 9 presents three types of masks worn by respondents. The results showed that most worn masks are non-woven facemasks, followed by cloth masks and surgical masks. The main reason for wearing non-woven facemasks is the availability of this type of mask produced by manufacturers mobilized by the Moroccan authorities, because access to surgical masks is almost impossible [52]. According to [53], the high demand for medical masks with the low price imposed by the administration has led to compromising their availability among pharmacists. However, about 28% of the studied samples were using surgical masks. As reported by [30], surgical masks are effective in reducing the spread of influenza virus as well as COVID-19 [54], but the misuse of facemasks could accelerate the propagation rate of COVID-19 amongst individuals because the virus may persist on the surface of the facemasks for days [55]. To be noted, polypropylene and polyester are the main raw materials used in the production of surgical masks [56]. Some of these materials may reach fresh water and then the ocean as a final acceptor in cases where the mask waste ends up in roads and landfill, which consequently will add more plastic and micro-plastic waste to the environment.

About 36% of surveyed individuals were using cloth masks; 13% were reusing their masks and 20% of respondents were washing them. A previous study has compared cloth masks with surgical masks and revealed that cloth masks were related to a greater risk of penetration of microorganisms and influenza-like illness (ILI) compared to no masks [57]. In this sense, in addition to the incorrect removal of facemasks, their reuse could cause self-contamination. Li et al. [58] have reported that during SARS, the observation suggested that the risk of infection could be higher if there was a misuse of masks such as double masking due to moisture, pathogen retention, and liquid diffusion. These effects could be linked to cloth masks [58].

The third part of our survey focused on community behavior and the management of PPE (mask and glove) waste. About 72% of people that answered the survey used facemasks, and more than 80% of the sample did not use gloves (Figure 9).

The comparison between the percentage of mask users and their management of the already used facemasks showed that almost all respondents threw their masks away after the first use in the trash bin of their houses (Figure 10). Accordingly, domiciliary waste will become hazardous as it is mixed with medical waste (gloves and facemasks), which could be a contagion if the discarded PPE is used by infected people with or without symptoms. Most (87%) of the surveyed members of the community strongly disagreed with throwing out their used masks in the street and/or nature.

In addition to the daily use of facemasks by the community, the framework of the new COVID-19 procedures also required companies and industrial units to provide protective facemasks to their workers to avoid the contamination of the working environment among employees; additionally, they are required to change the facemasks every 4 h [29]. At this rate, a huge quantity of disposal facemask are daily generated, which could have heavy effects on the environment and wildlife if this waste is mismanaged. According to the IMANOR [59], the maximal limit of heavy metal concentrations of the parts constituting the mask must be as follows (ppm): 1 for As; 0.1 for Cd; 2 for Cr; 0.038 for Hg; 1 for Pb; 30 for Sb; 4 for Co; 4 for Ni; and 40 for Cu. Thus, this new practice could pose a real threat to public health and the environment, because only 70% of urban municipality solid waste in 2008 has been collected and only about 10% was treated in environmentally and socially ways. Meanwhile, there were about 3500 waste pickers, among them rubbish pickers called “Mikhala” in the Moroccan dialect, of which 10% are children living on or near open dumpsites; there are about 300 uncontrolled dumpsites present in Morocco [60]. This population is the most vulnerable to the consequences of inadequate management of facemasks [61,62], especially because they often do not wear and misuse personal protective equipment such as gloves and facemasks which may put them at risk if infected masks or gloves are mixed with domicile waste [63]. A study conducted with waste pickers from South Sudan has proven that these people are at higher risk of developing hepatitis C, pulmonary diseases, and HIV due to contact with hazardous health waste [64]. Thus, domiciliary waste including facemasks, gloves, and other sharp items contaminated by COVID-19 disposed of by infected people undergoing treatment at home may pose a risk of contagion to the formal and informal waste pickers [65]. Moreover, [66] have reported that almost all waste generated by industrial activities is thrown out in uncontrolled dumpsites, along rivers, municipal landfills, or in abandoned quarries without any control or treatment. Accordingly, this waste becomes hazardous when it is mixed with medical items used by infected people without overt symptoms, which will pose risk to both public health and the environment.

Otherwise, 87% of the respondents strongly disagreed with throwing out the used masks and gloves in the streets, roads, and nature, and instead, they strongly agreed with throwing them out in their household bins. However, this does not imply that the whole Moroccan population behaves similarly. Thus, discarded PPE items in public places could have harmful impacts on the environment and cause clogs of pipe systems. Recently, a field investigation performed by the Mobile Area Water & Sewer system to investigate clogs through using cameras inside sewer pipes showed facemasks pulled out by workers, causing clogs [67]. Therefore, there is an urgent need to change the community behavior toward the management of PPE items and consider the management of domestic and industrial waste as it is mixed with hazardous waste.

### 4.3. Estimation of Potential Environmental Impact

The behavior of the Moroccan population toward the use of facemasks permits us also to estimate the environmental footprint based on the amount of each type of facemask worn. In acknowledgment of the authors, few studies have documented the environmental footprint of the facemasks. Klemes et al. [68] studied the energy and environmental footprints of this type of PPE in the United Kingdom (U.K.) based on the life cycle assessment (LCA), which considers the impacts throughout the whole life of the PPE, starting with the extraction of the material, through the manufacture, transporting, and use, and ending with the treatment and/or the disposal of the waste [68]. The authors considered that the greenhouse gas (GHG) footprint for a single-use mask manufactured mainly from polypropylene (PP) material is equivalent to 0.059 kgCO_2_ eq. per piece. In the case of reusable cloth masks, the GHG footprint is around 0.036 kgCO_2_ eq. by usage, considering machine washing. By using these values, we estimated the equivalent GHG footprint for the masks worn in the two studied regions in Morocco (Table 3). With these estimations, the highest contribution to the GHG footprint was linked to the transportation of PPE from China (the main PPE manufacturer), at about 74.3% [69]. Morocco is self-sufficient in non-woven masks. Thus, the values of footprint have been modified to reduce the transporting effects. A GHG footprint of 0.017 kgCO_2_/pcs eq. was considered for the non-woven type of masks. An annual climate change impact of 224 kT CO_2_ eq. was estimated for the two regions under investigation, while 640 kT CO_2_ eq. was valued for the whole country.

The increasing use of PPE in Morocco also has an environmental impact on energy consumption, specifically during the production of the equipment. The consumed energy for producing facemasks ranges between 0.000792 and 0.0342 kWh for each piece [68]. Assuming a mean value of 0.01 kWh by piece, the total energy consumption required to produce the needed facemask in Morocco is about 60,000 GWh per year (Table 4).

The potential energy recovery is another environmental parameter that provides the energy footprint. This parameter is based on the predicted energy produced by incineration treatment of the mask waste, which results from the combustion of the PP material. To estimate this energy, only the single-use masks, which are the most susceptible to incineration, are considered. A value of 0.04 MJ per piece was used in these calculations [68]. Table 4 shows the values of energy recovery calculated by using data related to the studied community behavior toward the use of masks. The results show that the potential energy recovery for the used facemasks during one year in Rabat-Salé-Kénitra, Casablanca-Settat, and the whole country are 23,298, 36,653, and 171,288 GJ, respectively.

Proper use and management of this kind of waste could help to reduce the quantity of disposed PPE in nature and minimize the resultant environmental impact associated with this usage. Without a filtration efficiency consideration, the use of cloth masks has shown a lower environmental footprint. Furthermore, the reduction in transporting equipment could be a source of reduction in this footprint.

## 5. Discussion

The COVID-19 pandemic and the Moroccan safety measures have caused an increase in demand for masks and gloves. However, this study shows strict respect for Moroccan safety directives (disinfection of clothes, hands, and items once they arrive at their houses) by the studied populations in Rabat-Salé-Kénitra and Casablanca-Settat regions. Additionally, this study showed that almost all the masks used were thrown in the house trash, while 36% of the studied populations reused and washed their masks (using the tissue masks). 

The results obtained from the survey revealed that this part of the community used three types of facemasks more than two times per day, and they threw them out after the first use in the house trash bin. Using masks at this pace with the absence of approaches, strategies, and public policies regarding the management of such waste would drastically increase the quantity of discarded PPE that could litter the gutters and cause blockages of waterways and flooding, as well as the transmission of viruses and infectious diseases.

The estimation of quantity and weight of facemasks disposed daily by the community allowed us to clearly understand the potential environmental impact that could result from the mismanagement of this waste. The results showed that disposed PPE items associated with COVID-19 can become a threat to the environment and human health; in the Casablanca-Settat region, it was estimated that when 70% of people used one mask daily, around 3,521,298 disposed facemasks would be generated, equivalent to 30,212 kg of waste produced daily. As the use of PPE items continues, their released waste can easily end up in landfill, rivers, and oceans, especially if this waste is mixed with household trash and thrown out in streets and roads.

The estimation of the environmental impact of the masks used in Morocco over one year showed that the increment of facemasks by COVID-19 produced an annual GHG footprint of about 640 kT CO_2_ and energy consumption of 60,000 GWh: an environmental footprint that clearly could be reduced with more efficient use behavior.

Furthermore, 87% of the surveyed community threw out their used PPE items in their household bins mixed with domestic waste, which may put informal and formal waste pickers at risk if these items are infected by COVID-19. These people work in unsafe and unhealthy conditions with poor hygiene, lack of access to PPE items, or misuse of these items, which means that the waste-pickers are the most exposed population to contracting coronavirus because of the mismanagement of medical items associated with COVID-19.

It is worth noting that a specific type of Moroccan population answered this exploratory survey, and it cannot be utilized to draw general conclusions about the behaviors of the whole Moroccan population regarding the use and disposal of protective equipment. The population that answered this survey was only persons with internet connection and access to social media channels. It is assumed that this population which, according to the World Bank, is 74.3% of the total [70], has a higher environmental consciousness. Therefore, the results could be overestimated. However, the results of this work are in agreement with other contemporary studies [46]. Consequently, further research is needed to investigate the way in which the Moroccan community uses and manages their used PPE items to establish efficient strategies and policies. Additionally, field and laboratory work are necessary to define and evaluate the long-term impact of facemask and glove disposal by the public on the environment. Moreover, a policy regarding the use of facemasks in public needs to be established, and a strategic plan to manage this solid waste must be developed.

Otherwise, producing reusable and eco-friendly facemasks appears to be an adequate solution in helping to minimize their potential impact on the environment and reduce the quantity of facemasks used and generated daily. Designing reusable and washable facemasks with a 3D printer could also help in decreasing the potential impacts of PPE items on the environment. Recycling PPE into biofuel, ethanol, steam, and electricity was proposed in previous studies to face the increasing quantity of PPE waste in nature, including rivers and oceans. These alternative solutions could succeed in solving the current problem associated with PPE waste, but we found that the procedure of collecting, selecting, and sorting out this kind of waste remains a big challenge, which is common to all kinds of solid waste in Morocco. Thus, changing public behavior towards the use and the management of plastic waste including facemasks and gloves, and installing efficient and functional waste management facilities in urban areas for the disposal of PPE items, will definitely help to reduce plastic waste during the pandemic and afterwards.

## 6. Conclusions

The use of personal protective equipment is a new culture for many countries in Europe, America, and Africa. The rapid spread of COVID-19 has led many countries around the world to recommend the use of facemasks. Some countries went even further by imposing the use of facemasks, such as Morocco. In this particular context, almost all the Moroccan population wear facemasks to protect themselves from infection by coronavirus.

The present study has investigated Moroccan community behavior toward the use of PPE and the potential environmental impact of misusing and mismanaging this kind of item. Accordingly, an efficient survey has been developed which enabled investigation into the usage behavior of PPE items by the habitants of two Moroccan regions: Rabat-Salé-Kénitra and Casablanca-Settat. The survey showed that 70% of the respondents disposed of their discarded masks and gloves in their household bins and/or in open dumps after the first use, whereas nearly 30% of them did not wear masks because they did not leave their homes during the lockdown.

Through our study, we indicate that more than five million (equivalent to 40,000 kg) facemasks could be daily generated and disposed of by 70% of the population living in Rabat-Salé-Kénitra and Casablanca-Settat regions, which presents 35% of the total engendered facemask waste in Morocco. Moreover, the results revealed that the mismanagement of facemasks could harm the environment by adding more plastic and microplastic debris to the terrestrial and marine ecosystems.

Millions of used facemasks are generated daily and managed in different ways (in streets, roads, gardens, household bins, and/or open dumps). Accordingly, the estimated greenhouse gas footprint was about 640 kT CO_2_ eq./year for the whole of Morocco, and the energy consumption related to the manufacturing of facemasks was about 60,000 GWh/year. These estimated data suggest that stakeholders should reconsider the resulting impact of the misuse and mismanagement of these items.

To conclude, the survey results and analysis showed that user behavior toward the usage and management of facemasks strongly influences the environmental impacts, while the fabrication phase contributed mostly to the investigated impact categories. 

Adopting eco-friendly tools to produce these items and raising awareness about the long-term impacts of discarding facemasks in nature and/or mixing them with household waste is vital to avoid the associated risk on human health and the environment. Developing a national plan to manage the increasing rate of disposed facemasks in public areas and providing functional waste management facilities in both rural and urban areas is required to reduce PPE-related waste and minimize their effects on the environment.

## Figures and Tables

**Figure 1 ijerph-18-04382-f001:**
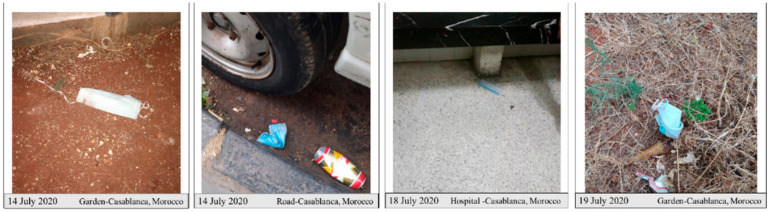
Pictures of discarded facemasks in roads and gardens, Casablanca, Morocco.

**Figure 2 ijerph-18-04382-f002:**
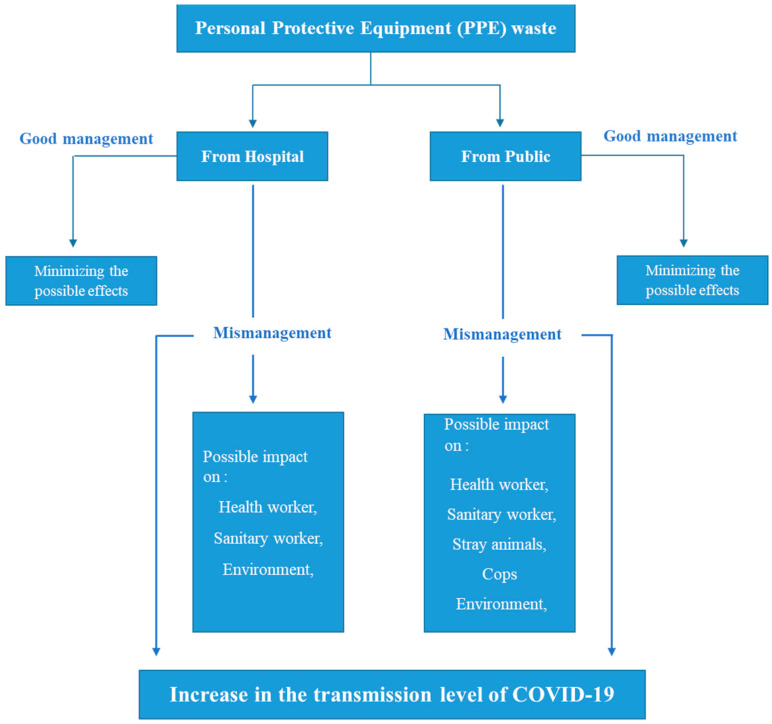
The possible impacts of mismanagement of personal protective equipment.

**Figure 3 ijerph-18-04382-f003:**
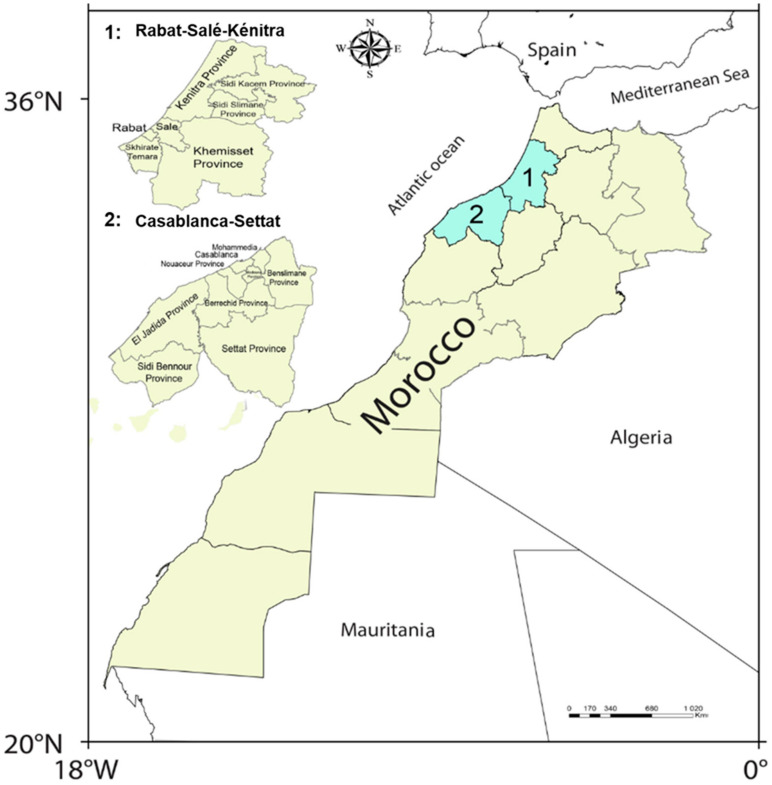
Rabat-Salé-Kénitra and Casablanca-Settat regions, Morocco, (location of studied cities).

**Figure 4 ijerph-18-04382-f004:**
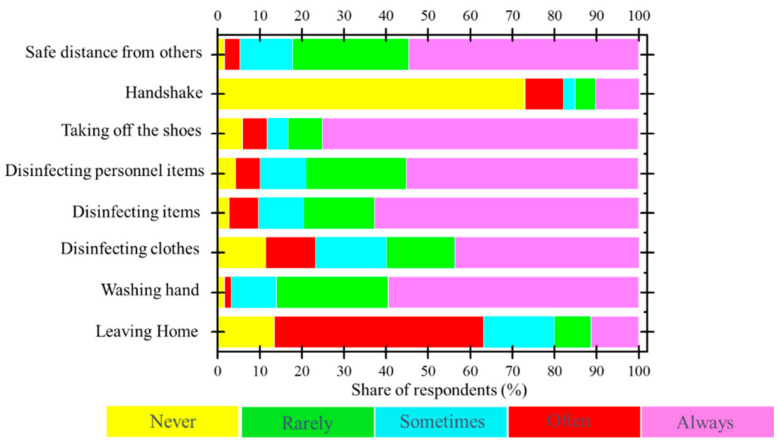
Habits and behaviors during the lockdown.

**Figure 5 ijerph-18-04382-f005:**
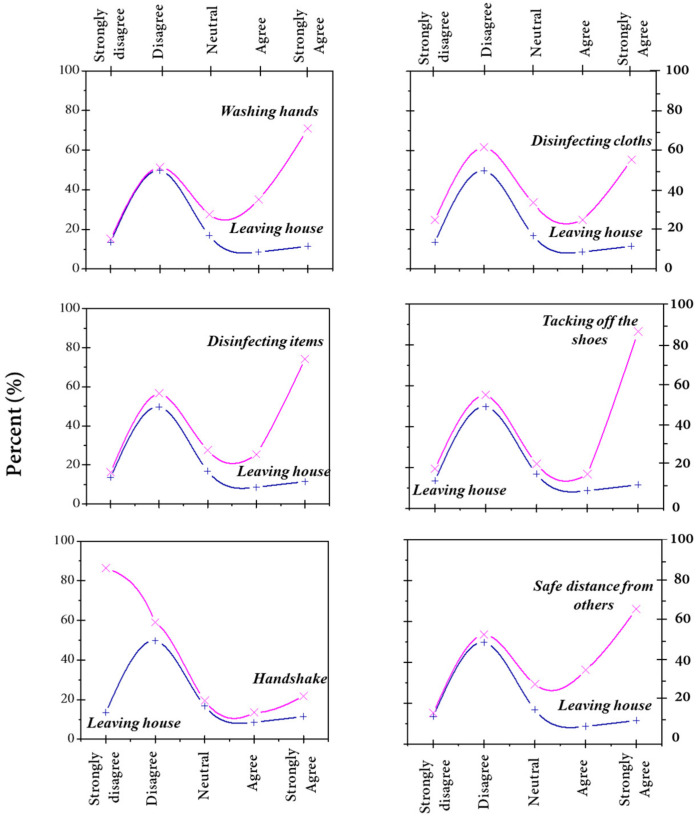
The relationships between leaving home and respecting sanitary measures.

**Figure 6 ijerph-18-04382-f006:**
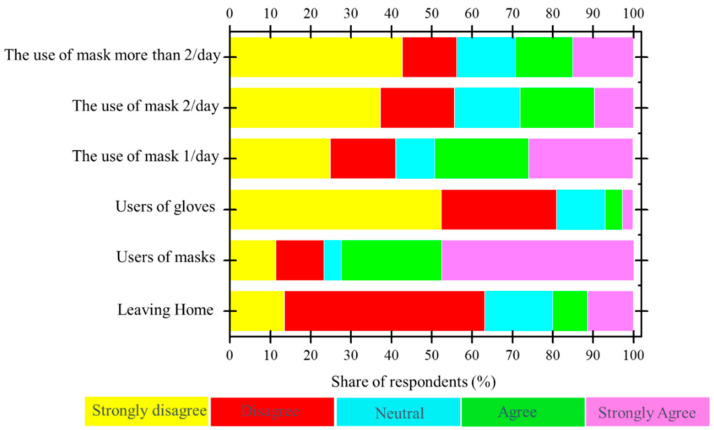
Frequency of the use of facemasks and gloves.

**Figure 7 ijerph-18-04382-f007:**
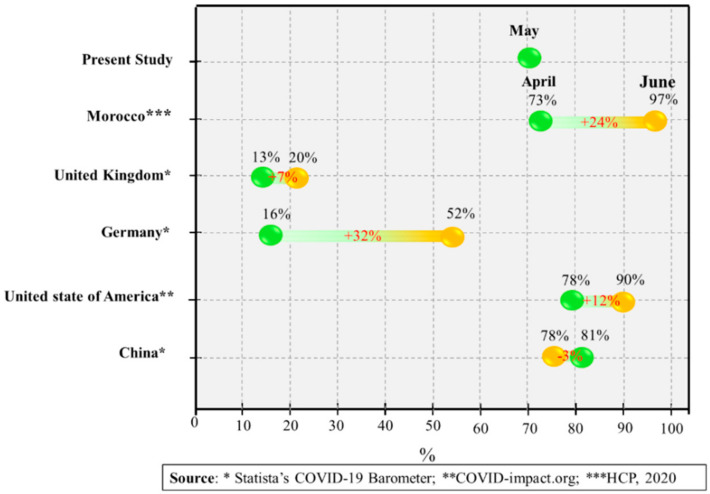
Masking up in a fight against COVID-19 between April and early June 2020 (results based on an online survey).

**Figure 8 ijerph-18-04382-f008:**
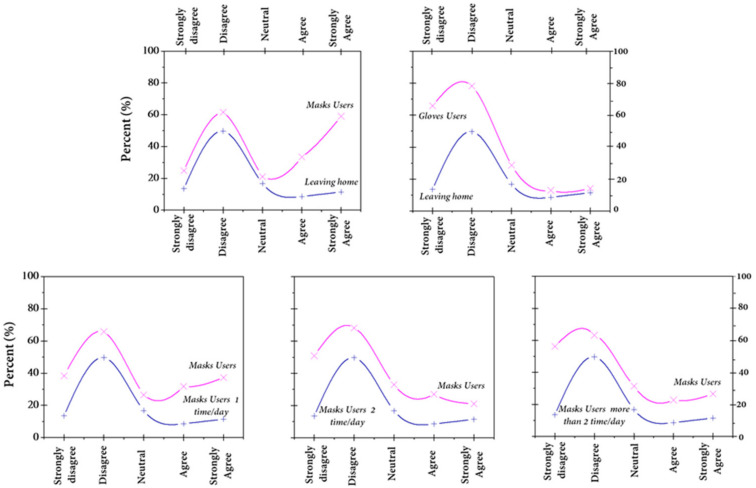
Relationship between leaving home and frequency of glove and mask use.

**Figure 9 ijerph-18-04382-f009:**
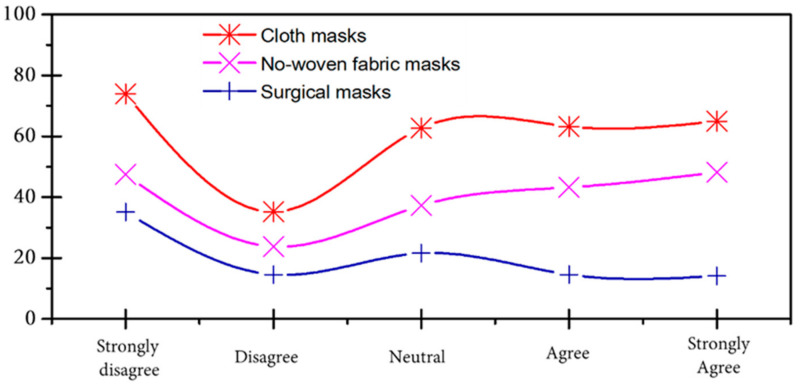
Comparison of the types of facemasks used.

**Figure 10 ijerph-18-04382-f010:**
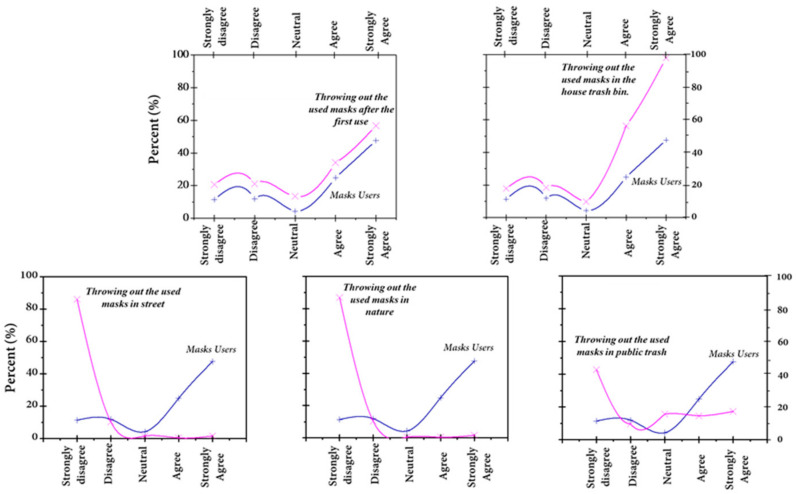
Management of discarded masks by the community.

**Table 1 ijerph-18-04382-t001:** Summary of the statistical parameters of this study.

Parameter	Value
Size of Sample Set	185
Confidence Level	90%
Margin of Error	7%
Cronbach’s alpha	0.6
KMO index	0.7
Bartlett’s Test of Sphericity	0.001

**Table 2 ijerph-18-04382-t002:** Estimated facemask disposal generation by Morocco; studied regions and surveyed population.

Present Study		**Population ***	**Urban Population (%) ***	**Rate (%) of Facemask Acceptance**	**Facemask/** **Capita**	**Estimated Daily Facemask Disposed**	**The Estimated Weight of Facemasks (kg)**
	Rabat-Salé-Kénitra Region	4,580,866	69.8	70 **	1	2,238,211	19,203
	Casablanca-Settat Region	6,842,255	73.52	70 **	1	3,521,298	30,212
	Surveyed Population	185	71.66	70 **	1	92.7997	0.7962
[48]	Morocco	36,913,924	64	70	1	16,537,438	140,568

* Source: HCP, 2014 [49]. ** % of the person using facemasks once, twice, or more per day according to the survey.

**Table 3 ijerph-18-04382-t003:** Annual GHG footprint calculated for the studied regions: Rabat-Salé-Kénitra and Casablanca-Settat, and Morocco according to the results of the present study.

	Number of Non-Woven Masks	Number of Surgical Masks	Number of Usage Cloth Masks	kg CO_2_ Eq. Non-Woven Masks	kg CO_2_ Eq. Surgical Masks	kg CO_2_ Eq. Cloth Masks	Total kg CO_2_ Eq.
Rabat-Salé-Kénitra Region	399,552,089	182,889,074	234,505,851	67,923,855	10,790,455	8,442,211	87,156,521
Casablanca-Settat Region	628,601,134	287,732,895	368,939,741	106,862,193	16,976,241	13,281,831	137,120,264
Morocco	2,937,580,638	1,344,634,199	1,724,130,263	499,388,708	79,333,418	62,068,689	640,790,816

**Table 4 ijerph-18-04382-t004:** Annual energy consumption and potential energy recovery for the studied regions: Rabat-Salé-Kénitra and Casablanca-Settat, and Morocco according to the results of the present study.

	Number of Masks	Number of Single-Use Masks	Energy Consumption (kWh)	Potential Energy Recovery MJ
Rabat-Salé-Kénitra Region	816,947,015	582,441,163	8,169,470	23,297,647
Casablanca-Settat Region	1,285,273,770	916,334,029	12,852,738	36,653,361
Morocco	6,006,345,100	4,282,214,834	60,063,451	171,288,593

## Data Availability

The data presented in this study are available on request from the corresponding author.

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
