# Peer review of "Disposal Behavior of Used Masks during the COVID-19 Pandemic in the Moroccan Community: Potential Environmental Impact"

_ijerph, 2021, doi:10.3390/ijerph18084382_

Round 1

Reviewer 1 Report

The title of the paper suggests a theme for a very timely scientific topic. However, what the paper offers is completely different from the title. Tittle suggests that it is about the environmental/public health impact made by the disposal of COVID19 masks. Then it is toned down in the abstract to an overview of management of used masks and gloves (i.e. waste). However, the paper is about neither. The paper only presents information gathered through a survey on mask/gloves disposal behavior of people from two regions. Disposal behavior is also important topic for a paper, but it is not scientific to call an apple an orange. Besides, the paper is not developed well. One example is how it ends abruptly: after presenting results, there is a recommendations and the flow stops there.

Reviewer 2 Report

Dear Authors,

I think that the manuscript submitted is interesting as a reports. The text presents an overview of masks and gloves waste management by the Moroccan population during the COVID-19 pandemic.  Among the objectives of the paper, there is the proposition of a national management strategy and a policy on the use of masks must be developed and shared with the whole population in order to minimize the risk of contamination from COVID-19, health vulnerability human and environmental degradation.

The manuscript falls into the category of brief communications or reports and not in original research.  The journal considers "original research manuscripts all papers carry scientifically valid experiments and provide a substantial amount of new information."
The same authors in the conclusions describe the further investigations necessary in the field and in the laboratory to better understand the impact of discarded masks and gloves on the environment.

The authors should review the conclusions, as much of it is dedicated to indicating the new investigations to be addressed in order to better understand the impact of discarded masks and gloves on the environment. 

  I think that the manuscript must be rewritten and provide not only anthropological-behavioral data.

Reviewer 3 Report

Comments

I have carefully reviewed the manuscript “COVID-19 Facemasks disposal: Potential impact on human  health and Environment (Morocco as a case study)”.  The paper was extremely difficult to read with poor use of tenses and inadequate sentence structure throughout the text. Issues identified in the manuscript are as follows;

  1. Abstract: English issues were abundant i.e. poor language structure, poor use of tenses etc. The authors are encouraged to revise the manuscript with the help of a native English speaker.

  1. Line 48-49. Please improve sentence.

  1. Line 50-52. English problems.

  1. Line 150-155.. Too many English issues exist in the current work. The text is very difficult to read

  1. I got exhausted identifying poor sentences. This work requires a complete overhaul.

  1. In the introduction statements regarding possible management approaches should be discussed. For instance a recent publication the conversion of waste N95 facemask  to energy and fuels via the thermochemical transformations of gasification hydrothermal liquefaction gasification–fermentation.  Why this approach may not work may be discussed.

  1. Methodology : Was there a reason for the choice of the categorizations employed in the survey. What do the authors mean by behavior and habits? Please provide some examples.

  1. When describing the terms in an equation, the terms should be exactly same. If italic terms are in the equation, the description should also contain terms in italics.

  1. The Authors must structure the paper in accordance with the template guide

  1. The core of this work (i.e. impact of face-mask on the environment) was only clearly specified in lines 314 to 336. I believe some modification of the title is required.

Round 2

Reviewer 1 Report

This version is better than the previous one. However, it seems that the authors have not fully captured the point I was trying to raise in my last round of comments. To appreciate the author’s persistence in improving the paper, I will get into a little more detail in this round.

The new title (and some other parts of the paper) is a 50% improvement as it now recognizes that the paper has something to do with the mask disposal behavior. However, the issue is about the other half of the title – the environmental impact part that comes after the colon. It is true that you have some data to discuss the mask disposal behavior and as a result you have a legitimate right to make some conclusions about it. However, the same is not true about the potential environmental impact. The data on behavior or your discussion on it, do not automatically reveal any new information about the environmental impact. My question is: based on what (new) data, are you trying to discuss or make conclusions on the environmental impact? You are basically putting the two things next to each other and tell the reader that it should be related. I am not denying the possibility to have an adverse impact on the environment, but when you are doing it without credible facts, it just becomes a speculation.

Here are a few things to think about:

  • You are implying that this 40,000kg of mask waste is a big amount/issue (I am just calling it mask waste, because very few uses gloves). However, you have not tried to put the numbers into perspective. Urban areas in Morocco produce about 0.76kg/capita amount of waste daily (* please see below for the source that I could found on internet, wihch is in line with typical numbers from many developing countries). This means that the 11.4million population in the study area (from your table 2) usually produces 8.7million kg of waste daily. So, the 40,000kg of daily mask waste is just a tiny fraction of the total daily waste, which is about  0.46%.

https://www.researchgate.net/publication/323175217_Management_of_Municipal_Solid_Waste_in_Morocco_The_Size_Effect_in_the_Distribution_of_Combustible_Components_and_Evaluation_of_the_Fuel_Fractions#:~:text=The%20generation%20of%20MSW%20in,Kg%20per%20capita%20per%20day).

  • If your numbers and the argument are correct, 70% people dispose the masks in their home trash bins. So, whatever is the normal waste collection/treatment mechanism used in these areas, that mechanism should take care of this 0.46% fraction too. If the current “normal” collection/treatment is not satisfactory, it is a separate issue about the whole waste management system, which has nothing to do with COVID19. However, still the fact is, proportionately, what kind of extra environmental damage can be caused by this extra 0.46%?

On the other hand, I found two important hints about some real concerns, that you have not elaborated enough (and also your survey data will not help either):

  • From the public health perspective: This is about the informal sector waste pickers, including children. I am not a medical expert, but as I understand, the real danger of possible exposure to COVID19 is within three days (assuming the spreading channel of touching contaminated waste). There is a good chance that the waste pickers handle domestic garbage (MSW) during this critical time. If they are not protecting themselves or if they are not knowledgeable about how COVID19 spreads, this is a loophole for a much larger adverse impact. Yet, this part in your paper is just limited to a sentence or two.
  • From the waste mismanagement perspective: The way I see it, the real issue is not about the masks that land in your trash bins, but the ones that do not. What you have shown in Figure 1 is a common problem on a global scale, irrespective of the labeling of countries based on developing versus developed. This new stream of waste in the public places can very well lead to clogging of drainage systems in the short run, but it will do a lot more harm to the whole environment in the long run. Although you have included a figure, your survey did not focus on this part of the problem.

Bottom line is that, you have a reasonably good data set to explain disposal behavior. Use it to make conclusions about the behavior, rather than trying to make unsubstantiated conclusions/recommendations about the environment.

Reviewer 2 Report

The authors have improved the manuscript using the suggestions.

I think that  the manuscript can be accepted in current form 

Author Response

Thank you very much by the positive comment. Authors think that part of the improvement was due to the valuable suggestions of the reviewer in the previous review. 

Reviewer 3 Report

I have carefully reviewed the manuscript “Disposal behavior of used masks during COVID-19 pandemic in the Moroccan community: Potential
Environmental impact”.  After my initial (previous review) comments, my final comment is as follows;

My queries have been resolved. I however suggest that assuming that 70 % of the population uses one mask per day may not be a reflection of reality. The authors should please highlight the limitations of this study in the text.

Round 3

Reviewer 1 Report

The manuscript has now reached a level that it may be published. 

Please make sure to;

1) Add a summary of your "conclusions" in a new separate section 5, while keeping section 4 only for the discussion.

2) Take some steps to improve the English language used in the paper.  There are still language issues here and there. 
